# Use of Antimicrobial Photodynamic Therapy to Inactivate Multidrug-Resistant *Klebsiella pneumoniae*: Scoping Review

**DOI:** 10.3390/pharmaceutics16121626

**Published:** 2024-12-23

**Authors:** Angélica R. Bravo, Felipe Alejandro Fuentealba, Iván A. González, Christian Erick Palavecino

**Affiliations:** 1Laboratorio de Microbiología Celular, Centro de Ciencias Médicas aplicadas, Facultad de Medicina y Ciencias de la Salud, Universidad Central de Chile, Lord Cochrane 418, Santiago 8330546, Chile; angelica.bravo@ucentral.cl (A.R.B.); felipe.fuentealba@alumnos.ucentral.cl (F.A.F.); 2Departamento de Química, Facultad de Ciencias Naturales, Matemáticas y del Medio Ambiente, Universidad Tecnológica Metropolitana, Las Palmeras 3360, Ñuñoa, Santiago 7800003, Chile; igonzalezp@utem.cl

**Keywords:** PRISMA-ScR, synergism with antibiotics, multidrug resistance

## Abstract

*Klebsiella pneumoniae* is a Gram-negative bacillus responsible for a wide variety of potentially fatal infections and, in turn, constitutes a critical agent of healthcare-associated infections. Moreover, *K. pneumoniae* is characterized by multi-drug-resistant (MDR) bacteria, such as extended-spectrum beta-lactamases (ESBL) and carbapenemase (KPC) producer strains, representing a significant health problem. Because resistances make it difficult to eradicate using antibiotics, antimicrobial photodynamic therapy (aPDT) promises to be a favorable approach to complementing conventional therapy against MDR bacteria. This study aims to provide relevant bibliographic information on the state of the art of application of aPDT against *K. pneumoniae* and MDR *K. pneumoniae*. Our methodology follows a protocol using the PRISMA extension for scoping reviews (PRISMA-ScR) guidelines, and the search consults the PubMed (MESH), Google Scholar, and Scopus databases from January 2012 to September 2024. The eligibility criteria were (1) original articles after 2012 referring to antimicrobial photodynamic activity in *K. pneumoniae* in vitro and in vivo: clinical applications and synergism with antibiotics, other antimicrobial drugs, or PS coupled to other particles, (2) articles in English, and (3) articles peer-reviewed. Results. Following two independent searches in databases, 298 records were found. After applying eligibility criteria and various filters, such as removing duplicates, 25 studies were included in this review. The evidence demonstrates the effectiveness of aPDT in vitro in eradicating sensitive or MDR-*K. pneumoniae* strains, including strains producing biofilms, ESBL, and KPC. Finally, it is concluded that aPDT is a recommended antimicrobial therapy, but more research in vivo is needed to support studies in humans.

## 1. Introduction

Enterobacteriaceae comprise 50% of clinical isolates in healthcare-associated infections (HAIs), and 80% are Gram-negative bacteria. Within the Enterobacteriaceae, the genus *Klebsiella* spp. is the second most important, and *Klebsiella pneumoniae* is the most prevalent species with the most significant clinical relevance [1]. *K. pneumoniae* is the cause of a substantial proportion of HAIs (30%), which includes soft tissue infections, urinary tract infections, pneumonia, and sepsis [2]. Due to multiple pathogenicity factors, such as adhesins and capsule formation, that act as an antiphagocytic factor, *K. pneumoniae* is considered an intrinsically pathogenic bacterium [3]. *K. pneumoniae* belongs to the group of bacteria called ESKAPE, which refers to six bacteria responsible for most nosocomial infections that “escape” the effects of antimicrobials and includes *Enterococcus faecium*, *Staphylococcus aureus*, *K. pneumoniae*, *Acinetobacter baumannii*, *Pseudomonas aeruginosa*, and *Enterobacter* spp. [3,4]. In-hospital strains of *K. pneumoniae* reach high levels of resistance to antimicrobials, reaching 61.4% of strains with multidrug resistance (MDR), 22% of strains with widespread antimicrobial resistance (XDR), and 1.8% of strains pan-resistant to all antimicrobials (PDR) [5]. All *K. pneumoniae* strains present natural resistance to ampicillin encoded, for example, in the BlaTEM-1, and many clinical isolates are producers of extended-spectrum beta-lactamases (ESBL) [6]. XDR bacteria may also produce carbapenemases, β-lactamases that hydrolyze penicillins, cephalosporins, carbapenems, and monobactams [7].

Carbapenamase-producing strains represent a severe risk since they are associated with infections with high morbidity and mortality, especially in patients with prolonged stays in the ICU or exposed to invasive devices [8,9]. These strains show resistance to all beta-lactams (penicillins, cephalosporins, monobactams, and generally intermediate resistance to carbapenems), so effective antimicrobial options are often lacking, and treatment depends typically on more toxic drugs such as aminoglycosides and polymyxins [10]. Strains resistant to polymyxin, such as colistin, are increasing and represent an alarming proportion of isolates in some health centers associated with higher mortality [7]. On the other hand, they have a large dissemination capacity because ESBL-type resistance is mediated by plasmids encoded in the bla_kpc_ gene [11]. Dissemination via plasmids is much faster, as evidenced by the fact that practically 70% of the strains collected by the US CDC, as well as in Greece, Israel, Norway, Brazil, and Argentina, are the exact clone, ST258, the hyper-epidemic clone of *Klebsiella pneumoniae* producing Bla_KPC-2_ [11]. Therefore, the antibiotics currently used are often ineffective in treating infections caused by multidrug-resistant bacteria. Consequently, developing non-antibiotic antimicrobial therapeutic alternatives is necessary [12]. Several new strategies have been developed, such as using metallic nanoparticles, cationic polymers, peptidoglycans, nanocarriers, photo thermotherapy, and photodynamic therapy [13]. Cationic compounds have been particularly widely used as PSs since they can weaken the permeability barrier of the outer membrane, allowing photosensitizer penetration [14].

Antimicrobial Photodynamic therapy (aPDT) has demonstrated a substantial antimicrobial potential among therapeutic alternatives. APDT, used against cancer cells and microorganisms, uses photosensitizer (PS) molecules activated by light, causing local oxidative stress [15]. It has also been used to treat various contaminations and for disinfection purposes, such as disinfection of water, blood, surfaces (hospitals), and medical devices, as well as food and crops [16]. As shown in Figure 1, PSs are non-toxic molecules that absorb energy at a specific wavelength (Figure 1A) and transfer it to molecular oxygen in biological solutions (Figure 1B) [13]. The activation of oxygen produces reactive oxygen species (ROS), such as superoxide (O_2_^•−^), hydrogen peroxide (H_2_O_2_), hydroxyl radicals (^•^OH), and singlet oxygen (^1^O_2_) (Figure 1C). ROS causes bacterial cell death by oxidizing organic macromolecules that are part of the cell envelope, such as lipids and proteins, or in the cytoplasm, such as nucleic acids. Bacterial cell death is unspecific because it destroys structures such as the plasma membrane, cell wall, and bacterial genome and can also distress vital processes by affecting enzymes and ribosomes (Figure 1D) [17]. The generation of ROS will be effective, depending on the photochemical characteristics of the photosensitizer; for example, a longer lifetime in excited states is essential to improve the probability of interacting with triple oxygen [18,19,20]. PS can be administered systemically intravenously into the bloodstream or applied locally as a topical to the skin [15]. Once applied, PS must be distributed into tissues or absorbed topically, getting in contact with cancerous or bacterial cells. Up to this point, PS remains inert but can be activated by light irradiation at a given wavelength to produce type I or type II ROS. Then, the aPDT will eliminate cancer or microbial cells through ROS formation and indirectly by stimulating the immune response [13].

One limitation of the aPDT is that the PS is activated only in those places where light can be accessed on the surface or coatings of internal organs, which can be illuminated with probes. How much light penetrates living tissues will depend on its wavelength: a general rule for visible and NIR light says the shorter the wavelength (<600 nm), the less penetration; the longer the wavelength (>800 nm), the greater the penetration. Therefore, blue light penetrates poorly into the skin tissues, accessing only the dermis, while red light penetrates better into the dermis without even reaching the hypodermis. Infrared radiation, in theory, would penetrate the hypodermis [21]. However, some lasers, such as erbium YAG or chromium YSGG, with wavelengths much higher than 800 nm (2940 and 2780 nm, respectively), show very low penetration capacities [22]. Shorter wavelengths, however, are much more energetic than long wavelengths and, therefore, much more capable of activating PS molecules. Light ranges between 650–850 nm are considered the most suitable for aPDT; thus, PSs activated within these ranges have the most potential for clinical use. The strong activation produced by sorter wavelength light can be helpful for the aPDT treatment of superficial injuries, such as surgical wound infections. Also, PS activated at <600 nm could be beneficial in decontaminating surfaces or clinical materials [23]. Other considerations must be taken when choosing a PS, such as its pharmacokinetics, which indicates that some accumulate in organs such as the liver, spleen, and kidneys [24]. Also, the elimination half-life of PS ranges between 12 and 19 h, highlighting the need to take precautions due to the possible phototoxicity of sunlight [25].

This scoping review aimed to conduct systematic research of evidence in this area to help researchers choose an aPDT appropriate to their needs and to summarize the development stage of aPDT applied to *K. pneumoniae*. The following research question was formulated: What is known from the literature about aPDT treatment of multidrug resistance *K. pneumoniae* infection, either in vitro or in vivo, and how many photosensitizers are used for this purpose?

## 2. Materials and Methods

Our methodology follows a protocol using the PRISMA extension for scoping reviews (PRISMA-ScR) guidelines, which is used to conduct a scoping review [26]. This type of review was selected because it allows for a systematic review to synthesize literature evidence, provide an overview of the available research evidence, and elucidate potential gaps in the existing knowledge. The study has been registered on the Open Science Framework platform (Protocol Registration DOI: https://doi.org/10.17605/OSF.IO/VKTDW).

### 2.1. Search Strategy and Eligibility Criteria

The search used a horizon of works no earlier than 2012; considering the PICO criterion, we agreed on 10 years and a buffer of 20%, 12 years old. Original articles were searched by FAF and ARB by consulting the PubMed (MESH), Google Scholar, and Scopus databases from January 2012 to September 2024. The search in these databases was carried out using the terms: “*Klebsiella pneumoniae*” AND “multidrug resistance” AND “photodynamic therapy” AND “coupled photosensitizer” OR “photosensitizer” AND “gram-negative” AND “ESKAPE”.

Studies were included if they meet: (1) original articles after 2012 referring to photodynamics associated with antimicrobial activity in *K. pneumoniae* in vitro and in vivo: clinical applications and synergism with antibiotics, other antimicrobial drugs, or PS coupled to other particles, (2) articles in English, (3) articles peer-reviewed. The exclusion criteria were: (1) Studies before 2012 or research not published in journals with an editorial board, (2) articles that were not accessed, (3) studies that did not declare the photosensitizer compound/concentration or light source/doses, (4) studies that included bacteria from the ESKAPE group but did not include *K. pneumoniae*, (5) studies without clinical significance (no significant *K. pneumoniae* growth reduction).

### 2.2. Information Sources and Search

All data were extracted solely from the databases mentioned above. In the Scopus database, in the Documents section, the Search within was “All fields”, and in the Search document, all terms mentioned above were added, one by one, in the given order. Then, the date ranged from “2012” to “Present” and Search. The “Article” button was selected in Document type, as well as “English” Language and “Final” Publication state. All open access was revised. In the first instance, keywords were searched in the title and abstract. The subsequent search was based on inclusion/exclusion criteria. Final documents were collected on EndNote 21. This information was used to eliminate duplicates.

### 2.3. Selection of Sources of Evidence and Data Charting Process

The articles were filtered by applying the abovementioned criteria in a PRISMA guideline that helped us better describe the systematic review (Figure 2). The PRISMA criteria applied were as follows: Two researchers (ARB and FAF) conducted searches independently, collecting information according to previously assigned search criteria. Once each researcher obtained the results, they were pooled and compared to eliminate duplicates using EndNote 21 citation manager software. The most recent research was conducted on 4 October 2024. The articles found were read and filtered, applying the inclusion and exclusion criteria. The following relevant characteristics were extracted independently by two reviewers: (1) the names of authors; (2) the year of publication; (3) the aim of the study; (4) methods applied to analyze the aPDT treatment; (5) photosensitizer used mentioning doses; (6) light source and doses; (7) range of bacterial growth reduction from results; (8) present and possible future applications, and (9) the clinical importance of the findings.

### 2.4. Data Items and Synthesis of the Results

Two researchers (ARB and FAF) summarized and reviewed the data with the senior reviewer (Christian Erick Palavecino (CEP)). Data extracted were summarized in a table (Table 1) and a narrative summary.

## 3. Results

### 3.1. Selection of Articles for Information Extraction

As shown in Figure 2, each researcher found 298 articles in the first instance. Google Scholar found 198, PubMed 12, and Scopus 88 articles. Of these, a total of 134 were repeated, leaving 164 articles remaining. Applying the inclusion and exclusion criteria, another 99 articles were removed, leaving 65. Other criteria, such as the availability of the complete article, incomplete information, not being according to the research, or needing clinical significance, eliminated 40 articles, leaving 25 articles for analysis. The data were extracted from the selected articles, processed, and grouped by association. A narrative summary was performed to extract the most essential information from each article, and a resume table (Table 1) and pie chart (Figure 3) were constructed to facilitate reading.

### 3.2. PSs Used for K. pneumoniae aPDT

#### 3.2.1. Methylene Blue Is the Most Common PS

A widely used PS for aPDT is methylene blue (MB), which, under light activation, can act as a strong oxidant capable of destroying target cells through cell damage, alteration of membrane permeability, and protein inactivation. MB has a broad absorption range, is also a cheap compound present in hospitals, and is approved by the US Food and Drug Administration (FDA) for intravenous use in humans, which is why it is the main PS used in aPDT [51]. However, some contraindications occur in pregnant women and psychiatric patients treated with serotonergics. MB is a dye that does not present toxicity and has a high quantum efficiency, producing singlet oxygen. Its absorption spectrum is in the red region, between 590 and 660 nm, with maximum absorption at 668 nm, and it is effective against both Gram-positive and Gram-negative bacteria. For example, the aPDT activity of MB has been widely demonstrated in planktonic cells and biofilm-forming cells of *Klebsiella pneumoniae* [52]. This dye may work at low concentrations; for example, as low as 10 mM of MB activated with 5 J/cm^2^ of LED light at 660 nm produced a 5 log_10_ reduction in MDR-*K. pneumoniae* shedding [29]. Also, 100 µM of MB with 16 J/cm^2^ of white light decreased the bacterial load by 3 log_10_ of carbapenemase-producing *K. pneumoniae* strains [35]. At 100 µM of MB, it is shown that *K. pneumoniae* becomes less tolerant to aPDT treatment [37]. Songsantiphap and colleagues (2022) worked with clinical isolates of MDR-*K. pneumoniae* from hospitalized patients in Thailand, studying MB-aPDT’s efficacy in various concentrations and with various red-light fluence [47]. The authors found a significant reduction in viable cells, over 2 log_10_, when using a red-light fluence of 40 J/cm^2^ combined with an MB concentration of 50 mg/L (156.32 µM) [47]. MB can also be used coupled with other molecules, such as dextran-coated gold nanoparticles (GNP_DEX_-ConA), that improve the efficiency and selectivity of MB by reducing the bacterial load by 97% [40]; or in synergy combining MB 1 mg/mL with antibiotics such as ceftriaxone 32 µg/mL, on MDR clinical isolates of *K. pneumoniae*, irradiated with 25 J/cm^2^ of LED light at 660 nm, decrease in 3.5 log_10_ the bacterial load [39]. Preclinical studies have also used MB for aPDT treatment of inflammatory deep tissue abscesses of purulent encapsulated lesions [43]. The authors first observed that planktonic cultures of *K. pneumoniae* required fluence of 25 J/cm^2^ (for 30 min) for approximately 7-fold log_10_ bacterial reduction with an MB concentration of 312.6 µM [43].

#### 3.2.2. Other PSs and Derivatives of PSs Active Against *K. pneumoniae*

Porfimer sodium or Photofrin (PF) is a PS activated by red light [23,53]. It is approved by the FDA to treat patients with certain types of esophageal, lung, bladder, cervix, and uterus cancer [54]. The neutral/anionic charge of PF makes it of low accessibility to negatively charged bacterial membranes. However, the PF may be mixed with carriers such as the nontoxic inorganic salt potassium iodide (KI), which helps to circumvent this disadvantage. For example, KI potentiates broad-spectrum antimicrobial photodynamic inactivation of PF against Gram-negative bacteria such as *K. pneumoniae* [36]. The authors showed that 100 mM KI + 10 mM PF activated with 415 nm light (10 J/cm^2^) eradicated > 6 log_10_ *K. pneumoniae* and other Gram-negative bacteria [36]. Other researchers must mix 100 mM KI with 10 µM PF to significantly (*p* < 0.001) inactivate 99.9889% *K. pneumoniae* and other bacteria when activated at 415 nm with a fluence of 10 J/cm^2^ [44]. Another investigation group used KI at the same concentration (100 mM) combined with 0.005% MB for photoinactivation of pathogens of the nasal cavity, namely, methicillin-resistant *Staphylococcus aureus*, antibiotic-resistant *Klebsiella pneumoniae*, multi-antibiotic-resistant *Pseudomonas aeruginosa*, *Candida* spp., and SARS-CoV-2 [50]. They found that microbial suspensions incubated with 0.005% MB + 100 mM KI formulation led to almost complete photoinactivation of bacteria or fungi (up to ~5 log_10_) when exposed to LED-emitted red light (~660  ±  25 nm), at a fluence of 10 J/cm^2^ for 3 to 5 min.

The effects of the photosensitizer Rose Bengal (RB) with a light-emitting diode (LED) were also investigated on Enterobacteriaceae, including *Escherichia coli*, *Enterobacter cloacae*, *Klebsiella oxytoca* and *Klebsiella pneumoniae*. Using a blue LED unit (460 nm) and RB at a concentration of 50 µmol/L, the reduction in 6.76 log_10_ was observed in *K. pneumoniae* specifically, demonstrating that Enterobacteriaceae strains studied were sensitive to photodynamic therapy with RB [38].

Aminolevulinic acid (ALA) is a medication that can be applied to the skin, such as the face or scalp, to treat actinic keratosis (AK). AK is a skin condition that can become cancer [55]. Derivatives of ALA are the 5-aminolevulinic acid (5-ALA) and the 5-aminolevulinic acid methyl ester (5-MAL), which are not in their own right PSs but are precursors for PSs, have been used to reduce the bacterial burden of *K. pneumoniae*. The authors used 10 mM 5-ALA activated with fluence 120 J/cm^2^ of 400–780 nm white light and reduced ESBL-producing *K. pneumoniae* strains by 3.20 log_10_ [32]. Cationic PS should interact closely with the negatively charged bacterial envelope, producing a more significant cytotoxic effect [49]. This is the case of PSIR-3 (a PS compound based on a polipyridinic Ir (III) complex). After light-activation, PSIR-3 inhibited 3 log_10_ (>99.9%) bacterial growth in a minimum dose of 4 µg/mL after 30 min of light exposure (fluence of 17 µW/cm^2^). The authors found in PSIR-3 a synergistic effect with imipenem, significantly increasing the bacterial inhibition of *K. pneumoniae* in 6 log_10_ [49]. Subsequently, Bustamente and her team revealed that the mode of action of PSIR-3 was through the activation of type II ROS (producing singlet oxygen), which upregulated *K. pneumoniae* genes related to damage to the bacterial envelope [28]. Also, PSIR-3 was used to treat virulent clinical isolates of *K. pneumoniae*, showing effective inhibition when used at 4 µg/mL after 30 min with light exposure (fluence of 17 µW/cm^2^) and even more effective when used in combination with 4 mg/L of Cefotaxime (3log_10_ without Cefotaxime and 6 log_10_ with it) [34].

Miretti and colleagues evaluated another cationic PS [14]. They described the compound ZnTM2,3PyPz (Zinc(II) tetramethyltetrapyridino [2,3-*b*:2′,3′-*g*:2″,3″-*l*:2‴,3‴-*q*] porphyrazinium methylsulfate), a PS base on phthalocyanine with strong absorption in the red visible light (700 nm), low aggregation tendency, photostability and a high single oxygen quantum yield [56]. ZnTM2,3PyPz, a water-soluble phthalocyanine with Zinc substituent, showed potent inactivation of both reference strain *E. coli* ATCC 25922 and a clinical isolate of *K. pneumoniae* Carbapenemase (KPC)-producing bacteria. After irradiation with visible light (constant power density of 20.5 mW/cm^2^), using 3 µM of ZnTM2,3PyPz, the viability of both KPC (30 min irradiation time) and *E. coli* (10 min irradiation) decreased by more than 99% [14]. These results are promising considering the difficulties in treating MDR bacteria such as KPC-producing *K. pneumoniae*.

Ruthenium (Ru)-based PS has proven to kill *K. pneumoniae* with bactericidal doses of 8 µg/mL PSRu-L2 and 4 µg/mL PSRu-L3. These doses inhibit bacterial growth by 3 log_10_ (>99.9%) and have a lethality of 30 min with an LED light dose of 0.612 J/cm^2^ [45]. Moreover, the authors mention a “remarkable synergistic effect” when complementing the aPDT with cefotaxime, increasing the bactericidal effect of both Ru-PS over 6 log_10_.

Riboflavin, as the vitamin B2, is being used in aPDT. In vitro assays were made in whole blood donated by healthy patients from a hospital in China [48]. Blood was infected with 1 × 10 log_4_ of four bacteria from the ESKAPE group, including *K. pneumoniae*, which authors called pan-drug-resistant *Klebsiella pneumoniae* (PDRKP). The authors placed the blood into small illumination bags (40 × 60 mm) made of a unique material called ethylene vinyl acetate copolymer, whose ultraviolet (UV) transmittance was above 99%. They injected Riboflavin at a final concentration of 260 µmol/L. Using treatment with combined UVA and UVB, at wavelengths of 365 nm and 308 nm, respectively, and a combined irradiation intensity of 15 mW/cm^2^, with the lowest dose of 18 J/cm^2^ for 20 min, the PDRKP inactivation rate was above 80%. They also used a high dose of UV light, 54 J/cm^2^, which showed bacterial inactivation of over 90% and red blood function. Although the 18 J/cm^2^ dose inhibited the proliferation activity of lymphocytes in whole blood, it was not deactivated. Finally, the authors proposed a recirculating system for treating infected blood from patients as a supplementary treatment mode when only antibiotics are insufficient [48].

### 3.3. Combined or Coupled PS Used Against K. pneumoniae

The efficacy of MB *per se* in aPDT has been well reported, as we previously reviewed in Section 3.2.1. However, MDR microorganism eradication has yet to be ultimately achieved. Pursuing this goal, Tosato and colleagues 2020 used MB at 2.5–10 μM alone or in combination with 6-carboxypterin (Cap) at 100 μM. The aPDT was performed with ultraviolet and visible light (UV-A at 300–400 nm), with a dose of 14.88 J/cm^2^. The aPDT shows a synergistic effect on eradicating an MDR *K. pneumoniae* strain with a similar impact with planktonic or biofilm-growing cells [30]. The authors further observed that the lethal action continues after treatment, eradicating microorganisms growing in the biofilm without irradiation [30].

PS TMPyP (5,10,15,20-tetrakis(*N*-methylpyriminium-4-yl) porphyrin) used in combination with silver nanoparticles (AgNPs) was investigated in this study by Malá and colleagues looking for synergistic antibacterial effect. They showed that the optimal combination of TMPyP and AgNPs was estimated as 1.56–25 µM for TMPyP and 3.38 mg/L for AgNPs in case of MRSA and 1.56–50 µM for TMPyP and 3.38 mg/L for AgNPs in case of ESBL-KP (*K. pneumoniae*) at 45 min incubation with TMPyP and fluence of 10 J/cm^2^ of light [33].

Other molecules may work better combined with peptides, such as those based on benzodiazole conjugated to pathogen-binding moieties (e.g., antimicrobial peptides). Then, after aPDT, singlet oxygen is preferentially generated into the bacterial cells [46]. Benzodiazoles are small, neutral, and conjugatable photodynamic scaffolds for preparing targeted PS (nitrobenzoselenadiazole PS-peptide) such as the PEP3. The authors indicated that treatment with 50 μM of PEP3 activated by 60 min LED illumination at 470 nm with a power density of 44 ± 6 mW/cm^2^ could kill several clinic-isolated bacteria, such as MDR *K. pneumoniae*. In those conditions, the PEP3 compounds significantly (*p* < 0.001) reduced the bacterial viability > 5 log_10_ of MDR-*K. pneumoniae* [46].

### 3.4. Uses of PSs Against K. pneumoniae

#### 3.4.1. In Vivo Studies of Photosensitizers

Although MB has been used as a healthy prospect and widely accepted in PS in vitro studies, few studies have used animal models to evaluate whether the same efficacy is observed in vivo assays. Grego and fellows treated viper stomatitis by aPDT using MB as PS [31]. Infectious stomatitis or “mouth rot” is one of the most diagnosed diseases in captive reptiles, characteristic of the infection of the oral mucosa and surrounding tissues with Gram-negative bacteria, fungi, and viruses (commonly isolated in the oral cavity of ill reptiles). The authors isolated *Klebsiella pneumoniae*, *Pseudomonas aeruginosa*, and *Escherichia coli* from lesions in this study. Treatment consisted of 1 mL of 0.01% MB aqueous solution applied directly into the lesions and irradiated with a diode laser at 660 nm wavelength, with a fluence of 280 J/cm^2^, 8 J, and 80 s per point, 100 mW, spot size 0.028 cm^2^. After 3 months of weekly treatment, snakes presented clinical improvements, such as reducing inflammatory signs and no signs of resistance to aPDT or recurrence of infection [31].

Few studies use animal models to evaluate photodynamic therapy in vivo to treat *K. pneumoniae* infections. One of the most recent studies used porphyrin-based photosensitizer to assess the decrease in bacterial load and edema caused by *K. pneumoniae* infection [41]. This LD4 photosensitizer was activated with a red laser (650 nm) at 6 J/cm^2^ fluency. In a rat model, the author observed that the treatment also alleviated the signs of inflammation with the diminution of inflammatory leukocytes and pro-inflammatory cytokines such as IL-6, IL-10, and TNF-α. Photodynamic therapy with LD4 could also effectively improve the survival of rats with pneumonia induced by *K. pneumonia* infection and protect the integrity of pulmonary epithelial cells [41].

#### 3.4.2. Photosensitizers for Environmental Improvement

Photodynamics can also be used to reduce the bacterial load of domestic water and facilitate the treatment of greywater [16]. The authors have used porphyrin tetrachloride TMPyP3 photosensitizer to treat municipal wastewater and prevent multidrug-resistant (MDR) strains from spreading to the environment and reaching drinking water [42]. They activated the PS by exposure to violet-blue light (VBL) at 394 nm with 20 mW/cm^2^ power density on MDR strains of *Klebsiella pneumoniae* and *K. pneumoniae* OXA-48, as well as *Pseudomonas aeruginosa*, in tap water and wastewater. They observed that although unactivated PS is bactericidal, this occurs at high concentrations close to 50 mM for tap water and between 12–100 mM for wastewater. The activity increases significantly when light activates, reducing the concentration required in both cases to approximately 1.5 mM of TMPyP3 to eradicate *K. pneumoniae* and *P. aeruginosa*. The drugs also showed an anti-adhesion effect at the same concentrations or even lower, suggesting it could be used to control biofilm [42].

## 4. Discussion

### 4.1. Summary of Evidence

This scoping review identified 25 studies addressing photodynamic therapy investigations associated with antimicrobial activity for *K. pneumoniae* in vitro and in vivo. Clinical applications and synergism with antibiotics, other antimicrobial drugs, or PS coupled to other particles between 2012 and 2024 were addressed. Our findings indicate that MB is the most widely used PS agent for aPDT *K. pneumoniae*. Most of those studies probed the effectiveness of MB using in vitro approaches [29,35,37,40,47]. We only found one article involving MB in an in vivo study on captivity snakes [31]. This last article had a significant impact as they observed that aPDT could treat infections in vivo [31]. As MB is the PS of choice, daily efforts are made to search for new options to improve efficacy, reduce adverse effects, and improve delivery methods. MB is used with antibiotics [39] or chemical compounds like KI [50]. Specifically, the study of Costa Magacho carries out an approach to an in vivo study in humans by demonstrating the effectiveness of a portable laser to treat intranasal infections by photoinactivation therapy [39]. Moreover, we found several studies using PSs made of cationic compounds and peptidoglycans on their own or combined with nanoparticles or polymers to enhance the inhibitory effect or reduce the survival of *K. pneumoniae* [14,28,34,45,46]. The nanoparticles as carriers may address some limitations shown by PS complexes, such as photobleaching, the need for proximity to the targets, low singlet oxygen quantum yield, or reduced dispersion in water [57]. Interestingly, an important feature emerges from the concentrations of MB used in the different studies. We observed concentrations ranging between 3 µM [58] and 300 µM [43] when using only MB. There does not appear to be a relationship between concentration and the light doses. A correlation with the application or if it is being used with another compound is not observed either.

Selecting the appropriate light source and dose is another essential feature for aPDT effectiveness. The choice will be determined based on the depth of the lesion, the optical properties of the tissue, and the absorption spectrum of the PS [59]. One of the limitations of aPDT is that the PS only becomes activated in places where light can be accessed. How much light penetrates living tissues will depend on its wavelength: the shorter the wavelength (<600 nm), the less penetration; on the contrary, the longer the wavelength (>800 nm), the more remarkable penetration [60]. For example, blue light is about 400 nm and is used for superficial cutaneous treatment (penetration depth is around 1–2 mm). In comparison, red light is about 650 nm and is used for more profound tissue treatment (penetration depth around 1 cm) [61]. In this last example, the absorption spectrum peak of MB is at 668–609 nm, meaning that MB absorbs at the red-light wavelength. Another characteristic to consider when choosing a photosensitizer is the oxygen availability in the tissue to treat, as ROS production requires molecular oxygen, which is limited in some conditions, such as cancer [62].

Regarding applications of aPDT against *K. pneumoniae*, we found the treatment of surfaces or clinical material [23,30], the use in medicine [48], and even in water disinfection [30,42]. Nevertheless, the most significant number of articles are related to the decrease in survival of clinical isolates of MDR *K. pneumoniae* [14,28,34,35,38,39,40,45,46,47,50].

### 4.2. Limitations of This Study

A limitation of this review was the inability of the reviewers to access some of the selected articles, highlighting the need to expand accessibility to scientific information. Also, 11 articles were left behind because of all the bacteria studied; *K. pneumoniae* was almost the only one that did not have any clinical significance/relevance; this is, it showed statistically no significant inhibition percentage when exposed to aPDT, which limited our results. We found this interesting because most of the articles referred to using clinical isolates, and therefore, more difficult to count with the genetic background. More studies are needed to clarify why this happened.

### 4.3. Conclusions

A systematic literature search was conducted to reunite all available information about the stage of development of aPDT applied to *K. pneumoniae*. The evidence described in this scoping review demonstrates the effectiveness of aPDT in eradicating *K. pneumoniae* multidrug-resistant strains, including strains producing Biofilms, ESBL, and KPC, with different PSs in different concentrations, light doses, and applications. Ongoing research focuses on improving aPDT efficacy, reducing side effects, and expanding clinical applications. In addition, new approaches are being investigated to generate or discover new PSs or use those already known in combination with others or with carriers as nanoparticles. Moreover, no relationships were found between concentrations used of MB neither with light doses nor with applications. This is an exciting feature for future research. Finally, aPDT is a valuable, non-invasive antimicrobial option for treating *K. pneumoniae* infections. Nevertheless, more research is necessary to support its use in human patients.

## Figures and Tables

**Figure 1 pharmaceutics-16-01626-f001:**
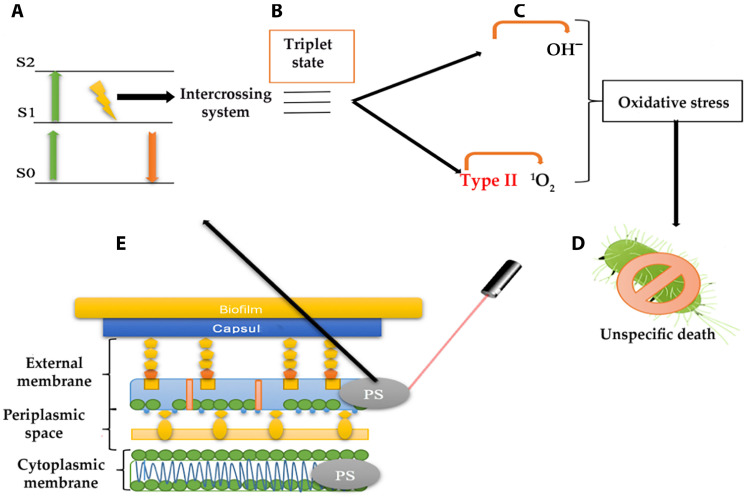
Scheme of the mechanism of action of photodynamic therapy. When the incident light interacts with the PS, it goes from the basal state S0 to an activated S1 or S2 in a singlet state (**A**). The excited e-spin can be reversed in the excited singlet and derived to an excited triplet state through an intersystem crossing process (**B**). In this state, the captured energy can be transferred to the nearby molecular oxygen accompanied by an e^−^ (Type I) or alone (Type II), generating OH^−^ or ^1^O_2_, respectively (**C**). The OH^−^ and ^1^O_2_ are reactive oxygen species (ROS) that produce photooxidative stress capable of destroying bacterial structures, causing unspecific death (**D**). A Gram-negative bacterial envelope’s representation, and interaction with a PS being irradiated by light is observed (**E**).

**Figure 2 pharmaceutics-16-01626-f002:**
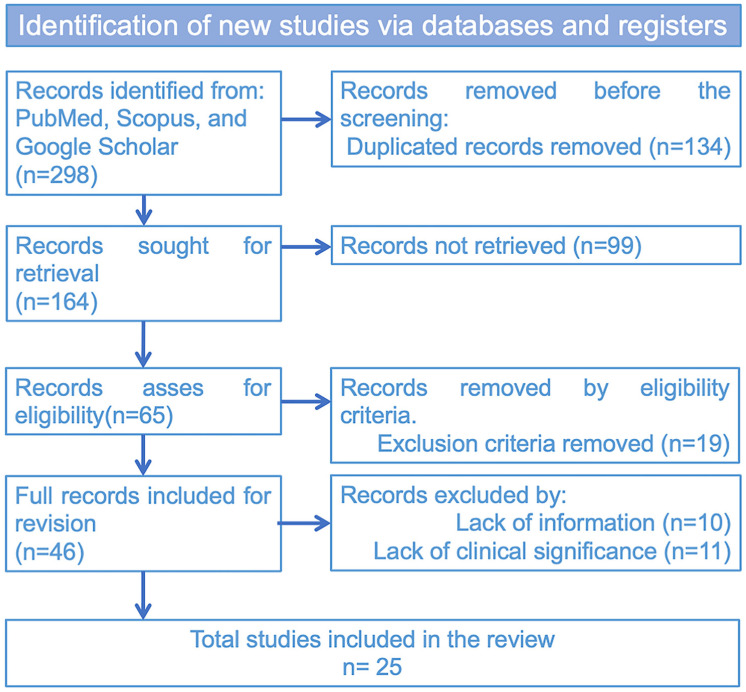
PRISMA flowchart showing the results of applying selection criteria to the bibliographic search.

**Figure 3 pharmaceutics-16-01626-f003:**
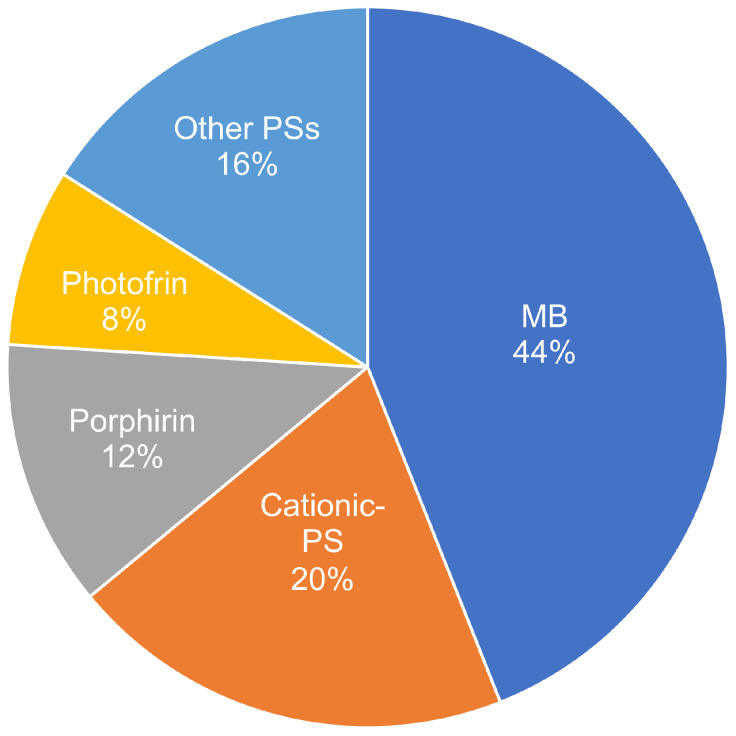
Diversity of PSs against *K. pneumoniae* aPDT. Pie distribution graphic of the diversity of compounds used as PSs in aPDT found against *K. pneumoniae*. The most common are Methylene Blue (MB) based (used alone or in combination with other compounds), with 44% of articles using it, followed by cationic-based PSs with 20%. Other PSs include derivatives of PSs and PSs fused to peptides or other compounds.

**Table 1 pharmaceutics-16-01626-t001:** List of articles and development of aPDT against *Klebsiella pneumoniae*.

Bacteria	PS/Dose	Dose/aPDT	Study/Reduction	Applications	Ref.
*K. pneumoniae* ATCC, ESBL and KPC	MB/3 μM	30 J/cm^2^ white halogen ligth at 640 nm	In vitro/5.9 log_10_ reduction.	For MDR bacteria	[27]
*K. pneumoniae* MDR	PSIR-3/4 μg/mL	0.017 mW/cm^2^ at white LED light	In vitro/3 log_10_ whitout imipenem and 6 log_10_ with imipenem	Clinical isolates, aPDT synergism with imipenem	[28]
*K. pneumoniae* MDR	MB/10 μM	white LED light, 600 nm/5 J/cm^2^	In vitro/5 lo_10_ reduction	Bring aPDT closer to implementation into mainstream medical practices.	[29]
*K. pneumoniae* MDR	Cap (100 μM) and MB (10 μM)	UV	In vitro/1.3 log_10_ Reduction	Biofilm producer MDR bacteria	[30]
*K. pneumoniae* MDR, *Morganella morganii*, *E. coli*, *P. aeruginosa* MDR.	MB 1 mL in aqueous solution at 0.01%	280 J/cm^2^ red laser at 600 nm	In vivo/100% reduction	MDR Clinical isolates in veterinary medicine	[31]
*K. pneumoniae* ESBL	5-ALA 10 mM and MAL 10 mM	120 J/cm^2^ white LED light at 400–780 nm.	In vitro/3.2 log_10_ for 5-ALA and 4.3 log_10_ for MAL	ESBL producers strains	[32]
*K. pneumoniae* ESBL	TMPyP 50 μM and AgNPS 3.38 mg/L	20 J/cm^2^ blue light at 421 nm	In vitro/3 log_10_ reduction	MDR bacteria, aPDT synergism with silver nanoparticles	[33]
*K. pneumoniae* KPC and *E. coli*	Zinc(II) ftalocianines ZnPC 18μM, TMAZnPC 6 μM and ZnTM 2.3 PyPz 3 μM	36.9 J/cm^2^ white LED light	In vitro/ - ZnPC: 0.7 log_10_ - TMAZnPC: 1.6 log_10_ - ZnRM 2.3 log_10_ - PyPz: 4.3 log_10_	MDR bacteria	[14]
*K. pneumoniae* ESBL and KPC	PSIR-3 (4 µg/mL) + Cefotaxime (4 mg/L)	0.017 mW/cm^2^ white LED light with 30-min of exposition	In vitro/3 log_10_ without cefotaxime and 6 log_10_ with cefotaxime	MDR clinical isolates, aPDT synergism with cefotaxime	[34]
*K. pneumoniae* ESBL and KPC	MB 100 μM	16 J/cm^2^ red LED light at 660 nm	In vitro/3 log_10_ reduction	MDR clinical isolates	[35]
*K. pneumoniae*, *S. aureus*, *Proteus mirabillis*, *P. aeruginosa* and *A. baumannii*	PF 10 μM + KI (100 mM)	10 J/cm^2^ at 415 nm (LED blue light)	In vitro/>6 log_10_ reduction in *Klebsiella pneumoniae*	Sensitive strains, aPDT synergism with KI	[36]
*K. pneumoniae*, *S. aureus*, *E. coli*, *P. aeruginosa* and *C. albicans*	MB 100 μM	100 mW/cm^2^ red light at 660 nm and 32 mW/cm^2^ blue light at 415 nm	In vitro/3 log_10_ reduction in *Klebsiella pneumoniae*	Antibiotic sensitive strains	[37]
*K. pneumoniae*, *K. oxytoca*, *Enterobacter cloacae*, and *E. coli*	RB 50 μmol/L	400 W blue LED light at 460 nm	In vitro/6.76 log_10_ for RB against *Klebsiella pneumoniae*.	Clinical isolates of the oral cavity	[38]
*K. pneumoniae* MDR, *E. coli* MDR and *Enterobacter aerogenes* MDR	MB 1 mg/mL + Ceftriaxone 300 μL	25 J/cm^2^ LED light at 660 nm ± 5 nm	In vitro/3.5 log_10_ reduction	Clinical isolates resistant to 3rd generation cephalosporins, aPDT synergism with ceftriaxone	[39]
*K. pneumoniae* MDR	MB 5 μg/mL	149.2 J/cm^2^ laser ligth at 522 nm	In vitro/> 3log_10_ reduction	Nosocomial clinical isolates, aPDT synergism with GNP_DEX_-ConA	[40]
*K. pneumoniae* (ATCC700603)	LD4 solution/180 μg/kg	6 J/cm^2^/red laser at 650 nm	In vivo (rat model of pneumonia)/7 log_10_ reduction	aPDT treatment of Acute pneumonia induced by *K. pneumoniae*	[41]
*K. pneumoniae* OXA-48	TMPyP3/1.562–6.25 μM	20 mW/cm^2^/violet-blue light at 394 nm	In vitro/values were reduced in half	aPDT treatment of tap water and municipal wastewater	[42]
*K. pneumoniae* ckp1	MB/100 μg/mL	25 J/cm^2/^665 nm (red light)	In vitro/7-fold log_10_ reduction in CFU.	They are intended for deep-tissue abscesses treatment.	[43]
*K. pneumoniae* and other SKAPE group bacteria	100 mmol/L KI with 10 μmol/L PF	10 J/cm^2^/415 nm (blue light)	In vitro/inactivate 99.9889% bacteria	Against bacteria of the SKAPE group.	[44]
*K. pneumoniae*	PSRu-L2 and PSRu-L3/8 µg/mL and 4 µg/mL, respectively	LED light dose of 0.612 J/cm^2^/450–460 nm	In vitro/3 log_10_ showing (>99.9%) inhibition of bacterial growth	*K. pneumoniae* treatment	[45]
*K. pneumoniae*.	50 μM of PEP3	LED white light dose at 44 ± 6 mW/cm^2^/470 nm	In vitro/>5 log_10_ reduction	Treatment of clinical isolates of *K. pneumoniae*.	[46]
*K. pneumoniae*.	MB 50 mg/L (156.32 µM)	Red-light fluence of 40 J/cm^2^/peak emission at 633 nm	In vitro/over 2 log_10_ reduction	Treatment of clinical isolates of *K. pneumoniae*.	[47]
*K. pneumoniae*, MRSA, *A. baumannii*, and ESBLs-producing *E. coli*	Riboflavin 260 µmol/L	UVA UVB combined dose of 18 J/cm^2^ for 20 min (wavelengths of 365 nm and 308 nm, respectively)	In vitro/over 80% bacterial inhibition	Generation of a recirculating system as a supplementary mode for treating patients infected with multiple ESAKPE bacteria, not responding to antibiotic treatment.	[48]
*K. penumoniae*	PSIR-3 4 µg/mL	0.017 mW/cm^2^ photon flux with LED white light	PSIR-3 compound inhibited 3 log_10_ (>99.9%) bacterial growth	efficiency of aPDT killing *K. pneumoniae* KPC + and − strains	[49]
MRSA, MDR-*P. aeruginosa*, a clinical isolate of *Pseudomonas* spp., and MDR-*K. pneumoniae*. A clinical isolate of *Candida* spp., and clinical isolates of SARS-CoV-2	0005% MB + 100 mM KI	red light-emitting LED 30 mW/cm^2^ (LED has emission at central wavelength of 655 nm), 3-min light exposure	In vitro/100% survival loss in all bacterial strains	Help in the design of an optimized protocol for future photoinactivation studies in clinical settings against board spectrum nasal pathogens.	[50]

Abbreviations: AgNPs: Silver nanoparticles; Cap: 6-carboxypterin; ESBL: extended-spectrum β-lactamases; GNP_DEX_-ConA: Dextran-coated gold nanoparticles targeted by Concavalin-A; KI: Iodinated potassium; KPC: carbapenemase-producing *Klebsiella pneumoniae*; MB: Methylene Blue; MDR: Multidrug-resistance; aPDT: Photodynamic Therapy; PF: Photofrin; PS: Photosensitizer; RB: Rose Bengal; TMPyP: Cationic porphyrin; ZnPC: Zinc Phthalocyanines.

## Data Availability

No new data were created or analyzed in this study. Data sharing is not applicable to this article.

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
