# Peer review of "Use of Antimicrobial Photodynamic Therapy to Inactivate Multidrug-Resistant Klebsiella pneumoniae: Scoping Review"

_pharmaceutics, 2024, doi:10.3390/pharmaceutics16121626_

Round 1

Reviewer 1 Report

Comments and Suggestions for Authors

Dear Authors,

The article " Use of Photodynamic Therapy to Inactivate Multidrug-resistant Klebsiella pneumoniae: Scoping Review" that was presented to me for evaluation is interesting.

As a reviewer, I propose the following changes:

Title:

in the title and throughout the manuscript I propose that, following custom, photodynamic therapy directed against, for example, bacteria be called aPDT (antimicrobial photodynamic therapy), to distinguish it from photodynamic therapy directed against cancer cells. Therefore, the title should be

Use of Antimicrobial Photodynamic Therapy to Inactivate Multidrug-resistant Klebsiella pneumoniae: Scoping Review

Affiliation:

Lines: 5-9 - remove the hyperlink in all author email addresses and add the ORCID number for those who have it.

Abstract

Lines 11-26 -   the abstract has a too long background (lines 11-19), but there is no important information from M&M, such as which databases were searched, how many records were initially found, how many were rejected, and why, and how many were included in the work. What was the time frame of the search, and so on?

I suggest you rewrite the abstract, reorganizing the text so that it maintains the proportions of the manuscript.

Keywords

I suggest for better searchability of the article change the given keywords because they are the same words that are in the title of the manuscript. Adding other words will increase the chance of finding the article by other researchers in the future.

Introduction

Line 32- should be Klebsiella (K.) pneumoniae - in the rest of the text you use the abbreviation K. pneumoniae which can be explained (expanded) here

Line 66 - you use the PSs shortcut, but the expansion of the shortcut is only on line 70.

Line 76 - I would also mention other important ROS not only OH+

Line 79 – also lists other intracellular structures, apart from those mentioned, that may be the target of ROS

Line 87 - I suggest replacing bacterial cells with microbial cells

Line 93 – “the longer the wavelength (>800 nm), the greater the penetration” –

This information should be corrected because it is true to some range. For example, Er:YAG laser 2940 nm is above 800 nm and the emitted electromagnetic wave penetrates very shallowly, only a few µm. It is similar for Er,Cr:YSGG laser (2780 nm) and CO2 laser (10600 nm)

Line 98 – is 650 -850 nm, should be 650 - 850 nm

Line 100 is PDT, should be aPDT

Lines 109- 111 Using PRISMA guidelines the focus question should be conducted following the PICO framework, as follows: (Population), (Intervention),  (Comparison), (Outcome) – please reorganize the question

Lines 112- 121 The presented Figure 1 is not needed in the manuscript - it is common knowledge. The manuscript is long anyway, so I suggest removing it before final publication.

Material and methods

Lines 130 - explain in the text why you adopted the lower time limit as of January 2012

Line 145 - provides the initials of the authors who conducted the search

Lines 160 – In the PRISMA flowchart, in the identification section, specify how many items were found in Pubmed, how many in Google Scholar or Scopus

Results

Line 183 - additionally provide the maximum absorption value for MB

Line 184 - is PDT, should be aPDT

Line 187 - adopt uniform nomenclature throughout the manuscript if you provide an energy density (fluence) value, e.g. 5 J/cm2  ( …, as low as 10 mM of MB activated with fluence 5 J/cm2 of LED light at 660 nm produced…)

Line 203 – (…for approximately 7-fold log10 bacterial reduction with an MB concentration of 312,6 µM for 30 minutes.) – 30 minutes is incubation time or irradiation time – explain  

Line 211 – is photofrin, should be Photofrin

Line 215 – is light dose, should be fluence

Line 220 – is 100mM, should be 100 mM

Line 222 - no fluence value was given in this task

Line 235 – is activated with 120 J cm2,should be - activated with fluence 120 J/cm2

Line 240 - in the described protocol and in other places in the manuscript describing aPDT, incubation time should be given as an important element of the aPDT algorithm.

Line 240  - is (17 µW/cm2 photon flux), should be – fluence 17 µW/cm2 photon flux

Line 244 – is K. Pneumoniae, should be K. pneumoniae

Line 247 - should be – fluence 17 µW/cm2

Line 252 – is 700nm, should be 700 nm

Line 256 – is (constant fluent of…), should be - constant power density of

Line 262 – should be - 30 min irradiation time

Line 269 – is (40mmx60mm), should be (40x60 mm)

Line 280 - is K. pneumoniae, should be K. pneumoniae (italics)

Lines 284 and 285 - is PDT, should be aPDT

Lines 293 and 294 – is 3.38 mg/l, should be 3.38 mg/L

Line 295 – is J/cm2, should be J/cm2

Line 303 – is (with 44 ± 6 mW/cm2), should be - with power density 44 ± 6 mW/cm2

Line 304 – is (p<0.001); Take one way of marking, e.g. on line 214 you write P<0.001

Line 312 - I suggest adding another very popular PS from this group which is TBO. If there is no such research, write why.

Line 313 – is K. pneumoniae, should be K. pneumoniae (italics)

Line 331 - with a near-infrared (NIR) laser (650 nm) at 6J /cm2 fluency.

              650 nm is a red light, not NIR!!! The same mistake is in Table 2 [32]

Line 332 – is 6J/cm2, should be 6 J/cm2

Line 334 - is PDT, should be aPDT

Line 343 – is (with 20 mW/cm2 power on MDR strains), should be with 20 mW/cm2 power density on MDR strains

Table 2

[28] – is 60 nm – it is a mistake

[52]-  is 1 ml, should be 1 mL

In column 3 (Dose/aPDT) - for easier understanding of readers, convert the power density units to one value. You have µW/cm2, mW/cm2

[48] is (of 365 and 308 respectively), should be - of 365 and 308 nm respectively

[43] is 4µg/mL, should be 4 µg/mL

Line 355 – is (TBO: met;), should be TBO: toluidine blue ortho

Discussion

Line 375 – is (between 3 μM and 0.3 mM), for easier understanding of readers, convert to one unit e.g. mM

Lines 392 and 400 – should be aPDT

The discussion is a bit short; it could be attempted to compare the ranges of laser/LED settings and PS concentrations for each of the discussed substances to show the lack of uniformity of protocols necessary to issue recommendations for clinical applications within evidence-based medicine

Author Response

Reviewer 1.

The article " Use of Photodynamic Therapy to Inactivate Multidrug-resistant Klebsiella pneumoniae: Scoping Review" that was presented to me for evaluation is interesting.

As a reviewer, I propose the following changes:

Title:

in the title and throughout the manuscript I propose that, following custom, photodynamic therapy directed against, for example, bacteria be called aPDT (antimicrobial photodynamic therapy), to distinguish it from photodynamic therapy directed against cancer cells. Therefore, the title should be

Use of Antimicrobial Photodynamic Therapy to Inactivate Multidrug-resistant Klebsiella pneumoniae: Scoping Review

Affiliation:

Answer: As requested by the reviewer, the title and the therapy name were changed in the article.

Use of antimicrobial Photodynamic Therapy to Inactivate Multidrug-resistant Klebsiella pneumoniae: Scoping Review

Lines: 5-9 - remove the hyperlink in all author email addresses and add the ORCID number for those who have it.

Answer: They are placed following the journal format.

Abstract

Lines 11-26 -   the abstract has a too long background (lines 11-19), but there is no important information from M&M, such as which databases were searched, how many records were initially found, how many were rejected, and why, and how many were included in the work. What was the time frame of the search, and so on?

I suggest you rewrite the abstract, reorganizing the text so that it maintains the proportions of the manuscript.

Answer: As requested by the reviewer, the abstract was revised and updated.

Keywords

I suggest for better searchability of the article change the given keywords because they are the same words that are in the title of the manuscript. Adding other words will increase the chance of finding the article by other researchers in the future.

Answer: As requested by the reviewer, the keywords were revised and updated.

Introduction

Line 32- should be Klebsiella (K.) pneumoniae- in the rest of the text you use the abbreviation K. pneumoniae, which can be explained (expanded) here.

Answer: As requested by the reviewer, the abbreviate was used in the rest of the text.

Line 66 - you use the PSs shortcut, but the expansion of the shortcut is only on line 70.

Answer. Yes, the expansion of the shortcut is located in the first place where the word appears.

Line 76 - I would also mention other important ROS not only OH+

Answer: As requested by the reviewer, the abbreviate was used in the rest of the text.

(ROS), such as superoxide (O2-), hydrogen peroxide (H2O2), hydroxyl radicals (OH), and singlet oxygen (1O2) (Fig 1C).

Line 79 – also lists other intracellular structures, apart from those mentioned, that may be the target of ROS

Answer: As requested by the reviewer, since prokaryotes do not have internal membranes, a derivation is made about possible effects on internal components of the bacterial cytoplasm that could be affected by aPDT is made.

…and can also distress vital processes by affecting enzymes and ribosomes.

Line 87 - I suggest replacing bacterial cells with microbial cells

Answer: As requested by the reviewer, the word was replaced.

…will eliminate cancer or microbial cells through ROS.

Line 93 – “the longer the wavelength (>800 nm), the greater the penetration” –

This information should be corrected because it is true to some range. For example, is above 800 nm and the emitted electromagnetic wave penetrates very shallowly, only a few µm. It is similar for Er,Cr:YSGG laser (2780 nm) and CO2 laser (10600 nm)

Answer: As requested by the reviewer, clarifying phrases were included.

a general rule for visible and NIR light says……

However, some lasers, such as erbium YAG or chromium YSGG, with wavelengths much higher than 800 nm (2940 and 2780 nm, respectively), show very low penetration capacities.

Line 98 – is 650 -850 nm, should be 650 - 850 nm

Answer: As requested by the reviewer, the text was updated.

Line 100 is PDT, should be aPDT

Answer: As requested by the reviewer, all were corrected.

Lines 109- 111 Using PRISMA guidelines the focus question should be conducted following the PICO framework, as follows: (Population), (Intervention),  (Comparison), (Outcome) – please reorganize the question

Answer: As requested by the reviewer, the question was reorganized.

Lines 112- 121 The presented Figure 1 is not needed in the manuscript - it is common knowledge. The manuscript is long anyway, so I suggest removing it before final publication.

Answer. This review is intended not only to be reviewed by people in the field of photodynamics but also by those who are looking for alternative therapies to conventional antibiotics and who know nothing about it. For this reason, we prefer to leave Figure 1, since it will be useful for those who are beginning their understanding of the therapy.

Material and methods

Lines 130 - explain in the text why you adopted the lower time limit as of January 2012.

Answer. Since the use of PDT as antimicrobial treatment is current. In accordance with the research question guidelines, the project sought to identify up-to-date research. Taking into account the PICO criterion, we agreed on 10 years and a buffer of 20%, that is, 12 years.

Line 145 - provides the initials of the authors who conducted the search

Answer. As requested by the reviewer, the initials of the authors who conducted the search were provided.

… articles were searched by FAF and ARB by consulting…

Lines 160 – In the PRISMA flowchart, in the identification section, specify how many items were found in Pubmed, how many in Google Scholar or Scopus

Answer. As requested by the reviewer, the text was included.

Results

Line 183 - additionally provide the maximum absorption value for MB

Answer. As requested by the reviewer, following paragraph is added: “with maximum absorption value at 668nm”

Line 184 - is PDT, should be aPDT

Answer. As requested by the reviewer, PDT was changed for aPDT

Line 187 - adopt uniform nomenclature throughout the manuscript if you provide an energy density (fluence) value, e.g. 5 J/cm2  ( …, as low as 10 mM of MB activated with fluence 5 J/cm2 of LED light at 660 nm produced…)

Line 203 – (…for approximately 7-fold log10 bacterial reduction with an MB concentration of 312,6 µM for 30 minutes.) – 30 minutes is incubation time or irradiation time – explain

Answer. As requested by the reviewer, the wording was changed.

Line 211 – is photofrin, should be Photofrin

Answer. As requested by the reviewer, “photofrin” was changed by PF

Line 215 – is light dose, should be fluence

Answer. As requested by the reviewer, the "light dose” was changed.

Line 220 – is 100mM, should be 100 mM

Answer. As requested by the reviewer, the change was carried out.

Line 222 - no fluence value was given in this task

Answer. As requested by the reviewer, fluence was given: “light fluence was 10 J/cm2

Line 235 – is activated with 120 J cm2,should be - activated with fluence 120 J/cm2

Answer. As requested by the reviewer, “fluence” is added

Line 240 - in the described protocol and in other places in the manuscript describing aPDT, incubation time should be given as an important element of the aPDT algorithm.

Answer. We understand that time is an important element of the algorithm; however is not one of the objectives of this search, as many of the research report the result as Joul/cm2, which includes the time used. Then, if a reader wants to know more information, they are invited to read the original paper.

Line 240  - is (17 µW/cm2 photon flux), should be – fluence 17 µW/cm2 photon flux

Answer. As requested by the reviewer, “fluence” was added

Line 244 – is K. Pneumoniae, should be K. Pneumoniae

Answer. As requested by the reviewer, the change was added

Line 247 - should be – fluence 17 µW/cm2

Answer. As requested by the reviewer, “fluence” is added

Line 252 – is 700nm, should be 700 nm

Answer. As requested by the reviewer, a change was added

Line 256 – is (constant fluent of…), should be - constant power density of

Answer. As requested by the reviewer, a change is added

Line 262 – should be - 30 min irradiation time

Answer. As requested by the reviewer, a change was added

Line 269 – is (40mmx60mm), should be (40x60 mm)

Answer. As requested by the reviewer, a change was added

Line 280 - is K. pneumoniae, should be K. pneumoniae (italics)

Answer. As requested by the reviewer, a change was added

Lines 284 and 285 - is PDT, should be aPDT

Answer. As requested by the reviewer, the change was added

Lines 293 and 294 – is 3.38 mg/l, should be 3.38 mg/L

Answer. As requested by the reviewer, a change was added

Line 295 – is J/cm2, should be J/cm2

Answer. As requested by the reviewer, change is added

Line 303 – is (with 44 ± 6 mW/cm2), should be - with power density 44 ± 6 mW/cm2

Answer. As requested by the reviewer, change is added

Line 304 – is (p<0.001); Take one way of marking, e.g. on line 214 you write P<0.001

Answer. As requested by the reviewer, change is added

Line 312 - I suggest adding another very popular PS from this group which is TBO. If there is no such research, write why.

Answer. Our search found no TBO as PS for K. pneumoniae.

Line 313 – is K. pneumoniae, should be K. pneumoniae (italics)

Answer. As requested by the reviewer, change is added

Line 331 - with a near-infrared (NIR) laser (650 nm) at 6J /cm2 fluency.

              650 nm is a red light, not NIR!!! The same mistake is in Table 2 [32]

Answer. As requested by the reviewer, change is added

Line 332 – is 6J/cm2, should be 6 J/cm2

Answer. As requested by the reviewer, change is added

Line 334 - is PDT, should be aPDT

Answer. As requested by the reviewer, change is added

Line 343 – is (with 20 mW/cm2 power on MDR strains), should be with 20 mW/cm2 power density on MDR strains

Answer. As requested by the reviewer, change is added

Table 2

[28] – is 60 nm – it is a mistake

Answer. As requested by the reviewer, change is added

[52]-  is 1 ml, should be 1 mL

Answer. As requested by the reviewer, change is added

In column 3 (Dose/aPDT) - for easier understanding of readers, convert the power density units to one value. You have µW/cm2, mW/cm2

Answer. We aimed to gather the information as published by the authors, without our intervention.

[48] is (of 365 and 308 respectively), should be - of 365 and 308 nm respectively

Answer. As requested by the reviewer, change is added

[43] is 4µg/mL, should be 4 µg/mL

Answer. As requested by the reviewer, change is added

Line 355 – is (TBO: met;), should be TBO: toluidine blue ortho

Answer. As requested by the reviewer, the table legend was amended.

Discussion

Line 375 – is (between 3 μM and 0.3 mM), for easier understanding of readers, convert to one unit e.g. mM

Answer. As requested by the reviewer, the text was changed.

Lines 392 and 400 – should be aPDT

  Answer. As requested by the reviewer, change is added

The discussion is a bit short; it could be attempted to compare the ranges of laser/LED settings and PS concentrations for each of the discussed substances to show the lack of uniformity of protocols necessary to issue recommendations for clinical applications within evidence-based medicine

Answer. As requested by the reviewer, the discussion section was improved

Reviewer 2 Report

Comments and Suggestions for Authors

I read with interest the paper titled "Use of Photodynamic Therapy to inactivate multidrug-resistant Klebsiella pneumoniae: Scoping review"

I have some questions that could perhaps, enhance the manuscript. 

1. Is the protocol for scoping review published elsewhere in an open repository for checking?

2. Line 124 - "This type of review was selected because it allows for a systematic review". Authors should clarify the difference between systematic and scoping reviews. If the objective is evidence, scoping review is fine, but for the systematic review some steps are missing. If the objective is to provide a broad overview of the topic, scoping is still fine. 

3. I wondering why authors used both EndNote and Mendeley and why. Please provide explanation. 

4. Organization of table 2 could be enhanced. Presenting the name of first author and year could be one of the options, instead of the reference column (or in addition). Eg: "Bravo et al. (2024) [99] ".

5. In table 1, How is the data organized? By year? By subgroups? By bacteria?

Author Response

Reviewer 2.

I read with interest the paper titled "Use of Photodynamic Therapy to inactivate multidrug-resistant Klebsiella pneumoniae: Scoping review"

I have some questions that could perhaps, enhance the manuscript. 

  1. Is the protocol for scoping review published elsewhere in an open repository for checking?

 Answer. Yes, it is: PRISMA extension for scoping reviews (PRISMA-ScR) guidelines (doi:10.7326/M18-0850)

  1. Line 124 - "This type of review was selected because it allows for a systematic review". Authors should clarify the difference between systematic and scoping reviews. If the objective is evidence, scoping review is fine, but for the systematic review some steps are missing. If the objective is to provide a broad overview of the topic, scoping is still fine. 

Answer. As requested by the reviewer, change is added

  1. I wondering why authors used both EndNote and Mendeley and why. Please provide explanation. 

Answer. Both were used to gather the information due to the familiarity of each researcher with distinct applications, considering that both allow lists importing and exporting compatible formats. The final document was done using just EndNote. This change was added.

  1. Organization of table 2 could be enhanced. Presenting the name of first author and year could be one of the options, instead of the reference column (or in addition). Eg: "Bravo et al. (2024) [99] ".

  Answer. Due to the format required by the journal, it is only left with the number and not the author's name. If the reviewer requests it, we will add the change.

  1. In table 1, How is the data organized? By year? By subgroups? By bacteria?

Answer. The table does not have a hierarchical order.

Reviewer 3 Report

Comments and Suggestions for Authors

The review article deals with the use of PDT as antimicrobial treatment for Klebsiella pneumoniae, as an alternative to antibiotics. The review article is novel, since use of PDT as antimicrobial treatment is a current hot topic, therefore, this review addresses the knowledge gap of the lack of appropriate review articles that have been published in this area till current date. References are adequate in number and they are recent. Some comments to be addressed are:

1- Line 110, the abbreviation "aPDT" needs to be clarified at the first instance for the readers. 

2- Since the authors referred to the conjugation of PS with nanparticles in the results section, the authors need to highlight the general merits of nanoparticles for loading PS in the discussion section.

3- Reference 13 is a review article published on the use of PDT for K. pneumoniae. Although the authors have referred to the reference in several instances in their manuscript, they didn't highlight the novelty/differences of their current manuscript compared to reference 13 (published in 2019). This needs to be addressed.

Author Response

Reviewer 3.

The review article deals with the use of PDT as antimicrobial treatment for Klebsiella pneumoniae, as an alternative to antibiotics. The review article is novel, since use of PDT as antimicrobial treatment is a current hot topic, therefore, this review addresses the knowledge gap of the lack of appropriate review articles that have been published in this area till current date. References are adequate in number and they are recent. Some comments to be addressed are:

1- Line 110, the abbreviation "aPDT" needs to be clarified at the first instance for the readers. 

Answer. As requested by the reviewer, change was added

2- Since the authors referred to the conjugation of PS with nanparticles in the results section, the authors need to highlight the general merits of nanoparticles for loading PS in the discussion section.

Answer. As requested by the reviewer, a paragraph was added in the discussion section.

The nanoparticles as carriers may address some limitations shown by PS complexes, such as photobleaching, the need for proximity to the targets, low singlet oxygen quantum yield, or reduced dispersion in water (57).

3- Reference 13 is a review article published on the use of PDT for K. pneumoniae. Although the authors have referred to the reference in several instances in their manuscript, they didn't highlight the novelty/differences of their current manuscript compared to reference 13 (published in 2019). This needs to be addressed.

Answer. Reference 13 is a comprehensive review that presents photodynamic as a viable option, mentioning the importance of the immune response and devices upon others; and uses K. pneumoniae to graph its antimicrobial utility, presenting few examples. The novelty of our work is a scoping review that focuses exclusively on K. pneumoniae aPDT, which allowed us to generate a table with the main studies and PS that have been tested on the bacteria.

Round 2

Reviewer 1 Report

Comments and Suggestions for Authors

Congratulations to the authors, the manuscript is well-corrected. 

Thank you for correcting most of the comments.

Tab 2 In column 3 (Dose/aPDT), convert the power density units to one value for readers to understand more easily. You have µW/cm2, mW/cm2. - this is an easy conversion - as you wrote, not all readers are experts, so I suggest changing the units to common ones, it will be unambiguous.

Author Response

Tab 2 In column 3 (Dose/aPDT), convert the power density units to one value so readers can u more easily. You have µW/cm2 and mW/cm2. This is an easy conversion. As you wrote, not all readers are experts, so I suggest changing the units to common ones; it will be unambiguous.

Answer: as requested by the reviewer, the power units in column 3, and table 2 were updated.
